# Tribological Performance of a Composite Cold Spray for Coated Bores

**Eduardo Tomanik [1,*], Laurent Aubanel [2], Michael Bussas [3], Francesco Delloro [2]** and **Thomas Lampke [3]**

1. Polytechnic School, Sao Paulo University, Sao Paulo 05508-010, Brazil
2. Mines Paris, MAT—Centre des Matériaux, PSL Research University, F-75005 Paris, France; laurent.aubanel@minesparis.psl.eu (L.A.); francesco.delloro@minesparis.psl.eu (F.D.)
3. Institute of Materials, Science and Engineering, Chemnitz University of Technology, D-09107 Chemnitz, Germany; michael.bussas@zeiss.com (M.B.); thomas.lampke@mb.tu-chemnitz.de (T.L.)
* Correspondence: eduardo.tomanik@usp.br

**Abstract:** The tribological performance of a thermal sprayed, mirror-like surface with localized protuberances was investigated through tribotests and computational simulation. A composite coating with a 410L steel matrix and M2 tool steel hard particles was applied by the cold spray process as a bore coating for combustion engines. The presence of protuberances promoted the quick formation of an antifriction tribofilm when tested with an SAE 0W-16 containing ZDDP and MoDTC, which significantly reduced the asperity friction in comparison to the conventional engine coated bores in reciprocating tribological tests. An in-house computational model using deterministic numerical methods was used for the mixed and hydrodynamic lubrication regime. Lubricant film thickness and friction were simulated for a piston ring versus the proposed coating. The computer simulations showed that the protuberances reduced the hydrodynamic friction by increasing the otherwise very thin oil film thickness of mirror-like surfaces. Although not intuitive, this result was caused by the reducing of the oil film shear rate.

**Keywords:** friction; surface texture; lubricant additive

## 1. Introduction

Engine friction losses represent one-third of passenger car fuel consumption and $CO_2$ emissions [1,2], around 50% of which are due to piston friction. The tribological system composed by the piston rings, the cylinder bore and the oil, as a reciprocating system with variable loads and temperatures, faces both boundary/mixed and hydrodynamic lubrication regimes. Indeed, the piston-bore sliding speed varies all along the stroke and depends on the engine regime (rpms). For a given rpm, the stroke can be divided in three parts: two zones near the top and bottom dead centers, where the speed is low and the regime is boundary/mixed, and the middle part of the stroke, where the speed is higher and the regime is hydrodynamic. At different rpms, the same division is still valid, but the extension of the three zones varies; at higher rpms, the hydrodynamic zone is larger than at lower rpms. As friction losses occur only after combustion, the corresponding 10% of fuel energy represents approx. 25% of fuel consumption. Combustion efficiency is ~40%, and the 10% (fuel energy) of friction losses occurs only after combustion, so in terms of fuel consumption, friction losses are 10/40 = 25%. See details in [3].

The main research paths to friction reduction include [4]:
- The use of materials/coatings with a low friction coefficient;
- Surface finish and/or texturing, especially by optimization using computer models and detailed topographic characterizations;
- Low-viscosity oils and improved friction-modifying additives.

In a lubricated tribological contact, even in a hydrodynamic regime in which the parts are not directly in contact, the surface texture is very important to promote good tribological

performance. The use of a surface with dimples has been widely studied for that purpose. When the void ratio created by the dimples is high, the coefficient of friction generally increases under boundary and mixed regimes due to higher contact stress [5]. On the other hand, in a hydrodynamic regime, the surface with dimples tested in [6] reduced the friction coefficient by up to 35% thanks to a lower oil shear rate.

Oil formulation composition has a major influence on friction properties. First, the viscosity plays an important role. Oils with high viscosity values reduce friction and wear in the boundary and mixed regimes but increase them in hydrodynamic regimes [7]. For this reason, low-viscosity oils are increasingly used in actual engines. The wear resistance in the mixed/boundary regime zones of the stroke is assured in this case by oil additives. Zinc dialkyldithiophosphate (ZDDP) has been widely used and studied as an antiwear additive. During friction, it reacts to create a thick tribofilm, protecting the surface from wear. As a drawback, it increases the coefficient of friction [8]. Molybdenum dialkyldithiocarbamate (MoDTC) is a well-known friction modifier that reacts in the contact areas to form $MoS_2$ solid lubricant. $MoS_2$ reduces the coefficient of friction but not the wear [9]. When ZDDP and MoDTC are combined, both a low friction coefficient and good wear protection can be obtained [10].

Fully formulated engine oils containing ZDDP and MoDTC among their additives show an interesting interplay with surface textures. Additives generally react first on asperities and then accumulate inside dimples or grooves to form a continuous protective film [11,12].

Different thermal spray technologies have been used to apply coatings onto engine cylinder bores, to reduce the interbore distance, to allow for easy cooling and to reduce friction losses using mirror-like surfaces. In comparison with cast materials, thermal sprayed coatings are characterized by a non-uniform microstructure due to the presence of pores and oxides, also inducing a non-homogenous hardness. The manufacture of these coatings, including two-way interactions of thermal spray and mechanical finishing, is subject to challenges with respect to effective and efficient processes [13,14].

### 1.1. Mirror-like Coated Bores Produced by Thermal Spraying

Mirror-like coated bores used in recent combustion engines are much smoother than gray cast iron blocks or iron cast sleeves on aluminum blocks. See Figures 1 and 2. As an example, Figure 2 illustrates the topography of two engine cylinder bores: a plateau-honed gray cast iron bore and a mirror-like coated bore. The second is significantly smoother than the first, with a Sa value eight times lower than that of the former. Due to their very low roughness, mirror-like coated bores are very effective in reducing engine friction losses at low and moderate sliding speeds. However, as the sliding speed increases and hydrodynamic losses dominate, as shown in Figure 3, such benefit is reduced and becomes negligible at high rpms.

A composite coating made with a 410L stainless steel matrix and hard M2 tool steel reinforcements was applied by the cold spray process. Two commercially available steel powders produced by Sandvik Osprey Ltd. were mixed in a volume ratio of 80/20, respectively. Their particle size distribution, as provided by the supplier, was +32–10 μm. The cold spray equipment used was 5/11 from Impact Innovations GmbH, equipped with an internal diameter coating device with nitrogen as the principal gas, a pressure of 5 MPa and a temperature of 900 °C. The stand-off distance and powder feeder rotation speed were set at 5 mm and 8 rpm, respectively. They were sprayed to form the composite coating directly onto 10 mm-thick laminated aluminum 6060 plates with dimensions of 65 by 40 mm.

Thanks to the local inhomogeneity of the coating hardness induced by the presence of M2 steel particles, the surface finishing resulted in a textured topography whereby the harder M2 particles became protuberant. It was then demonstrated that friction on that surface with protuberances promoted the formation of a thick tribofilm all around the protuberances, leading to a very low coefficient of friction (<0.04) and negligeable wear in the mixed/boundary regime. In the present article, the tribological behavior of the

same surface with protuberances was studied under a hydrodynamic regime. To do so, a deterministic computer model was applied. A reciprocating computer model was then used to predict the piston ring friction losses.

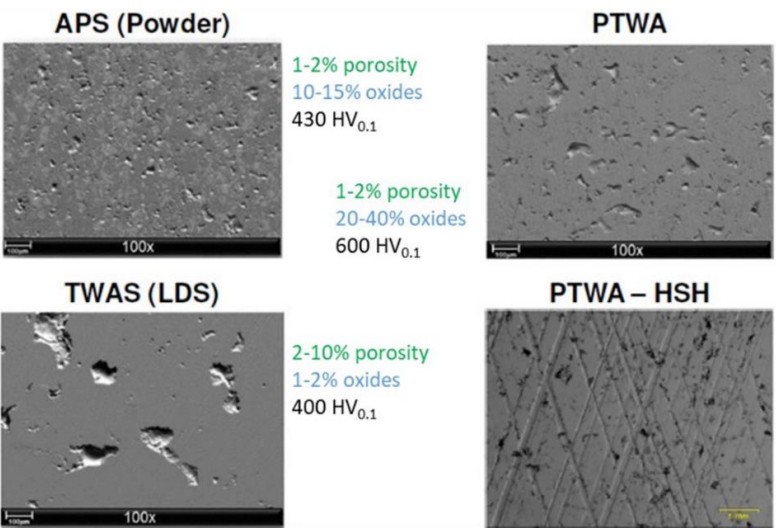

**Figure 1.** Some of the coated bore variants in production. Adapted from [15].

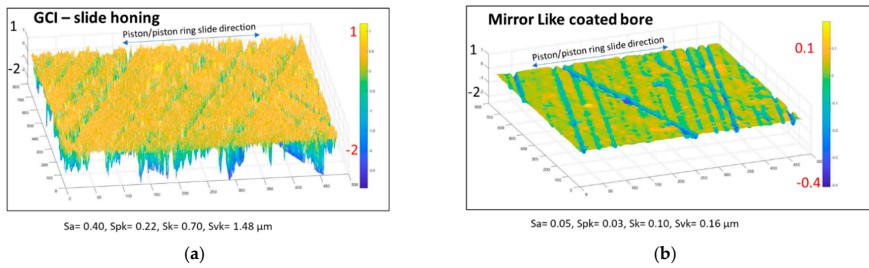

**Figure 2.** Topography and roughness of two production engine cylinder bores: (**a**) gray cast iron, plateau-honed bore; (**b**) mirror-like coated bore. Adapted from [15].

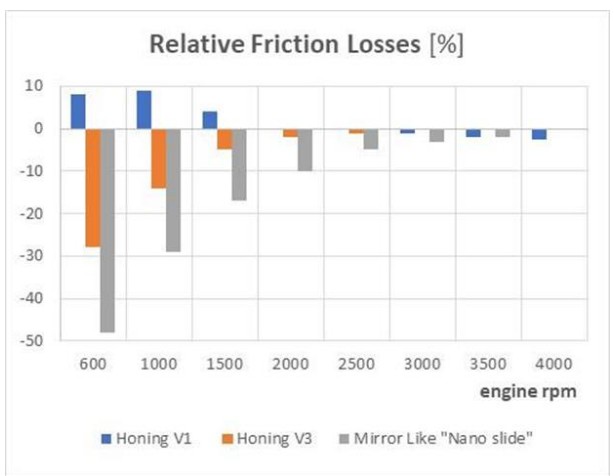

**Figure 3.** Relative friction losses of different engine cylinder bores. Adapted from [16].

## 1.2. Topographical Characterization of Textured Surfaces

The use of localized texturing creates the need for more complex surface quantification. A mirror-like surface with localized texturing cannot be described only by the usual statistical roughness parameters such as Sa, Spk, etc. Mirror-like and textured surfaces are mostly composed of a very smooth topography, with a few relevant topographic features. Special care is also need for waviness filtering. Few but deep localized features can evade

simpler waviness filters. Therefore, classical statistically based models may fail to reproduce the actual application, as discussed in [17]. Several standards deal with this statistical bias. For surfaces with localized features such as dimples or protuberances the norm DIN EN ISO 8785: 10/1999 suggests that surface imperfections that often represent a minor portion of the surface (<<20%) can be used as a reference. According to DIN EN ISO 21920-2:12/2022, all the 31 local surface imperfections described in DIN EN ISO 8785 are to be explicitly excluded from the amplitude calculation, Abott and hybrid parameters of DIN EN ISO 25178-2: 02/2020. This needs to be considered when calculating Sa, Spk, Sk and Svk to avoid outliers (local phenomena) adulterating the meaning of the roughness parameters. However, excluding the localized features results in a dilemma with respect to how to properly characterize and quantify, for example, dimples and protuberances. This topic is outside the scope of the current paper but is briefly discussed in Appendix A and in more detail in [14].

In the present study, a 13Mn6 LDS coating, which is currently applied in the production of engines, was used as a reference case for comparison. This coating shows upstanding material as described in DE 10 2012 002 766 B4 as Deckel-Ölvolumen ("cap oil volume") [18] (hereafter called protuberance) at comparatively high volumetric expansions. This kind of upstanding material also is observed around pores and grooves in the finishing process of similar specimens, along with unwelcomed phenomena of seizure in engine tests [19]. The origin of the surface material integrity issues can be explained by spray and layer build-up processes (large, porous holes and lower intermetallic connections). The relatively less impact of the spray quenching process in cold-sprayed finished 410L seems to result in only small pores and homogeneous finishing abrasion without upstanding material. The 20% M2 addition in the third sample results in a comparatively finer and more even distribution of protuberances (micro plateaus) compared to the 13Mn6LDS due to the surface finishing process. Indeed, the hardness difference between the 410L and the reinforcing M2 particles results in different abrasion rates so that in this case, the quantity of upstanding material is controlled by material composition.

### 1.3. Computer Models for Piston Ring and Cylinder Bores

As mentioned before, as the major source of engine friction, the piston system is one of the main subjects for the development of low-friction technologies. As a reciprocating system, the piston assembly can operate in boundary, mixed and hydrodynamic lubrication regimes. The hydrodynamic regime is more relevant for power losses and fuel consumption since it generally occurs at higher speeds. Figure 4 summarizes different approaches for simulating the lubrication behavior of a piston ring–cylinder bore system [17].

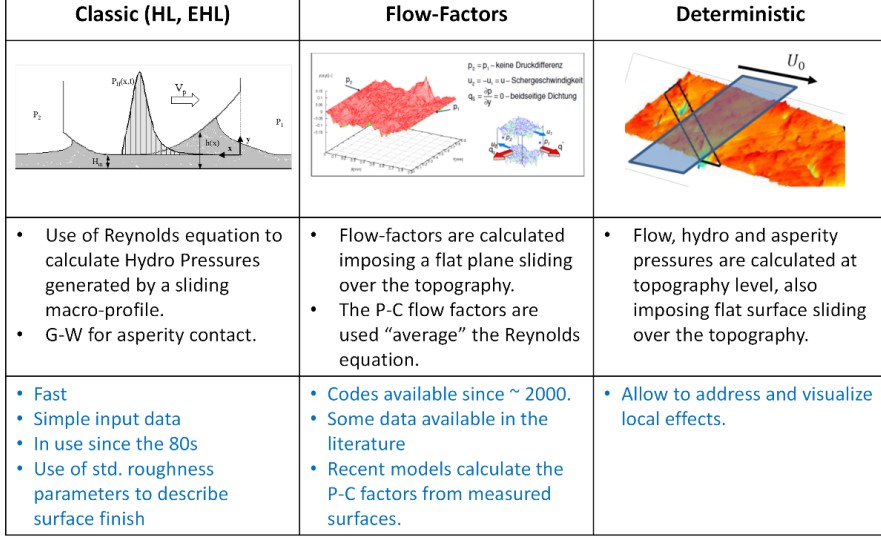

| Classic (HL, EHL) | Flow-Factors | Deterministic |
| --- | --- | --- |
| • Use of Reynolds equation to calculate Hydro Pressures generated by a sliding macro-profile.<br>• G-W for asperity contact. | • Flow-factors are calculated imposing a flat plane sliding over the topography.<br>• The P-C flow factors are used "average" the Reynolds equation. | • Flow, hydro and asperity pressures are calculated at topography level, also imposing flat surface sliding over the topography. |
| • Fast<br>• Simple input data<br>• In use since the 80s<br>• Use of std. roughness parameters to describe surface finish | • Codes available since ~ 2000.<br>• Some data available in the literature<br>• Recent models calculate the P-C factors from measured surfaces. | • Allow to address and visualize local effects. |

**Figure 4.** Computer models for a mixed lubricating regime. Adapted from [17].

- Classical Reynolds equation and stochastic asperity contact models: The hydrodynamic pressures are calculated using the Reynolds equation, considering the ring kinematics, the running profile and the physical properties of the lubricant. Under mixed lubrication conditions, where the oil film thickness is not enough to completely separate the surfaces, a statistically based rough contact model (usually the Greenwood–Williamson or Greenwood–Tripp model) is adopted to calculate the asperity contact pressures. This approach fails to account for the hydrodynamic pressures generated by the surface roughness, and it predicts zero hydrodynamic pressures for flat, parallel surfaces (e.g., oil control ring outer lands).

- Average Reynolds equation and flow factors: The previous approach can be improved for mixed lubrication analysis by incorporating the effect of surface roughness on lubrication through the adoption of averaging flow methods. In these cases, the influence of roughness (microscale) is considered by utilizing averaging parameters (flow factors) introduced in the Reynolds equation, which is then solved considering the macroscopic geometry of the contacting surfaces. The most common averaging method used for piston ring simulation is the Patir–Cheng average flow model, which provides a modified average Reynolds equation.

- Deterministic simulations: In this approach, the effect of surface topography is directly considered for the simulation of a mixed lubrication regime. The hydrodynamic and asperity contact problems are solved simultaneously in the same numerical framework. Due to the high CPU efforts, the most common approach is to segment the surface topography into small slices and then solve the coupled hydrodynamic and asperity contact problems in a quasistatic manner by imposing different separations between each slice and a rigid plane. Alternatively, fully deterministic simulations can be employed such that no segmentation is applied beforehand, and the surface separations are calculated based on the instantaneous load equilibrium.

In the current work, we investigated the mixed and hydrodynamic regimes of different mirror-like coated surfaces used for engine cylinder liners. The novel surface is one obtained by composite cold spray, whereby the powder hard particles produced localized protuberances after surface finishing. Such protuberances helped to increase the oil film, reducing friction losses at higher speeds. An improved contact model to approximate the regions in contact areas was used in a deterministic model in order to better represent the relatively large contact in the protuberances. The calculated deterministic results were then used in a reciprocating simulation to mimic the behavior of a piston ring. When combined with empirical friction coefficient values obtained with fully formulated oils containing friction modifiers, the novel composite cold spray surface obtained the lowest friction for both boundary and hydrodynamic lubricant regimes.

## 2. Materials and Methods

The effects of material content, coating process and lube oil FM additive on friction were investigated by tribological tests and computational simulations as described below.

*Coated Cylinder Bores*

Three different coatings were investigated in this work:

- LDS: already in production in recent engines; made with 13Mn6 steel and obtained by the LDS (Lichtbogen–Draht–Spritzen) process, also known as BSC (bore spray coating), TWA (twin wire arc) or TWAS (twin wire arc spray);
- 410L: obtained by a cold spray process using a 410L feed stock powder alone;
- 410L + 20% M2: obtained by cold spraying using the same 410L powder with the addition of 20% vol. M2 tool steel powder. The composite coating obtained by cold spraying the powder mixture consisted of a 410L stainless steel matrix with a hardness of 356 ± 39 Hv0.01. The 20% in volume of M2 tool steel in the powder stock resulted in about 8% area of particles with a hardness of 799 ± 81 Hv0.01.

All coatings underwent surface finishing before tribological testing. SEM cross sections and top views are shown in Figure 5.

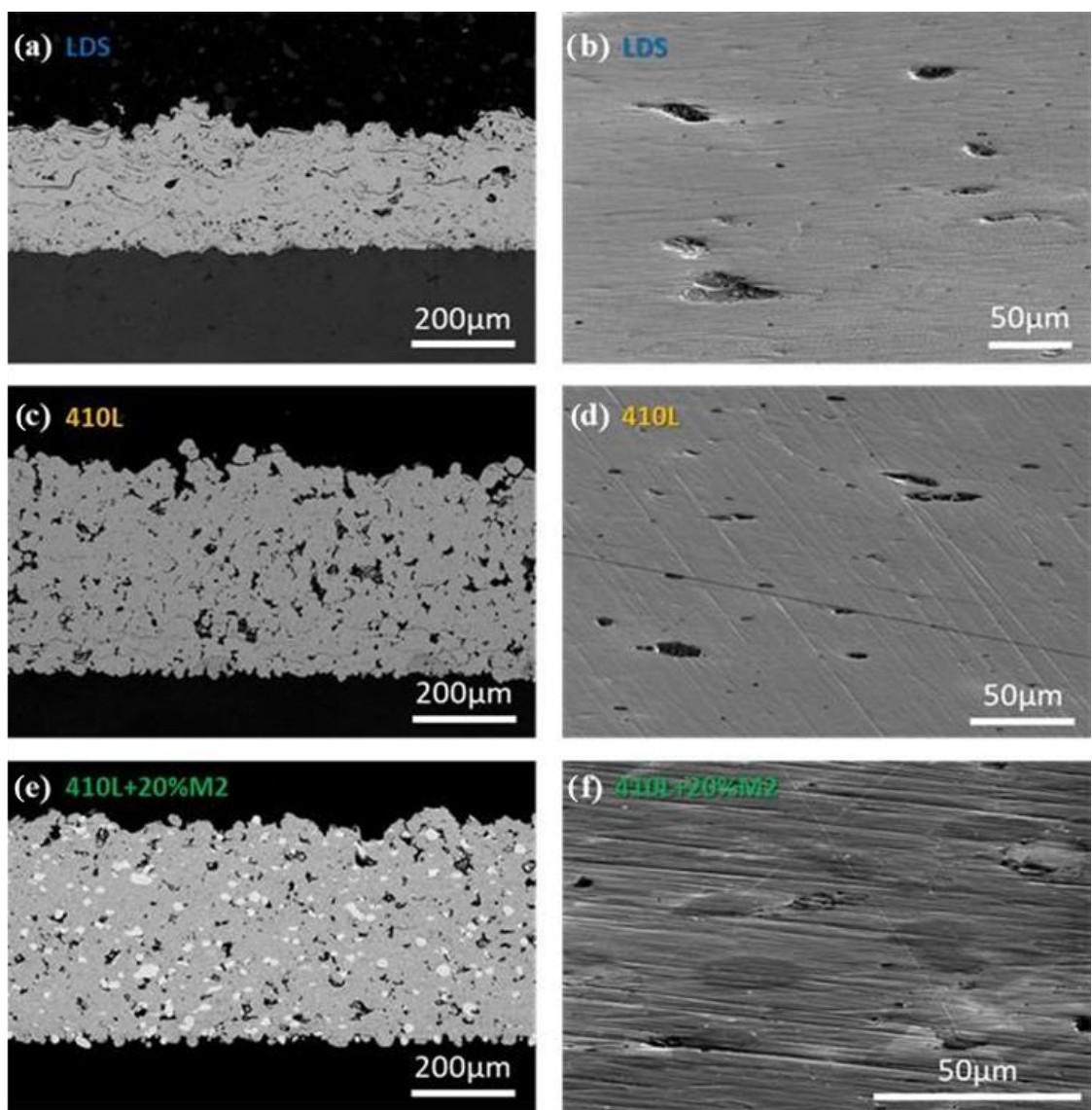

**Figure 5.** SEM cross-sectional (left column) and tilted top views (right) of the three coatings studied: (**a**,**b**) 13Mn6 LDS coating; (**c**,**d**) 410L cold spray coating; (**e**,**f**) 410L + 20% vol. M2 cold spray coating.

The surface finish for the three coatings was made by polishing with 1200 SiC abrasive paper using a standard metallographic preparation polish machine with water as lubricant and cleaning fluid. After different trials, this procedure was chosen because it resulted in surface textures close to those obtained by honing, which is the process traditionally used in industrial engine production. The hardness difference inside the composite coating naturally induced a specific topography, whereby the hard particles were raised $119 \pm 15$ nm above the matrix, creating protuberances. This is reflected in a higher Spk for the composite coating. Figure 6 shows the topography measured by white light interferometry (WLI) for the three coatings after surface finishing. The "grooves" were caused by the relatively rough polishing paper. The presence of some localized burrs (e.g., at x, y = 800, 100; 800, 500; 1100, 400) in the LDS can also be observed, which resulted in low-density but relatively large values of protuberance (Table 1).

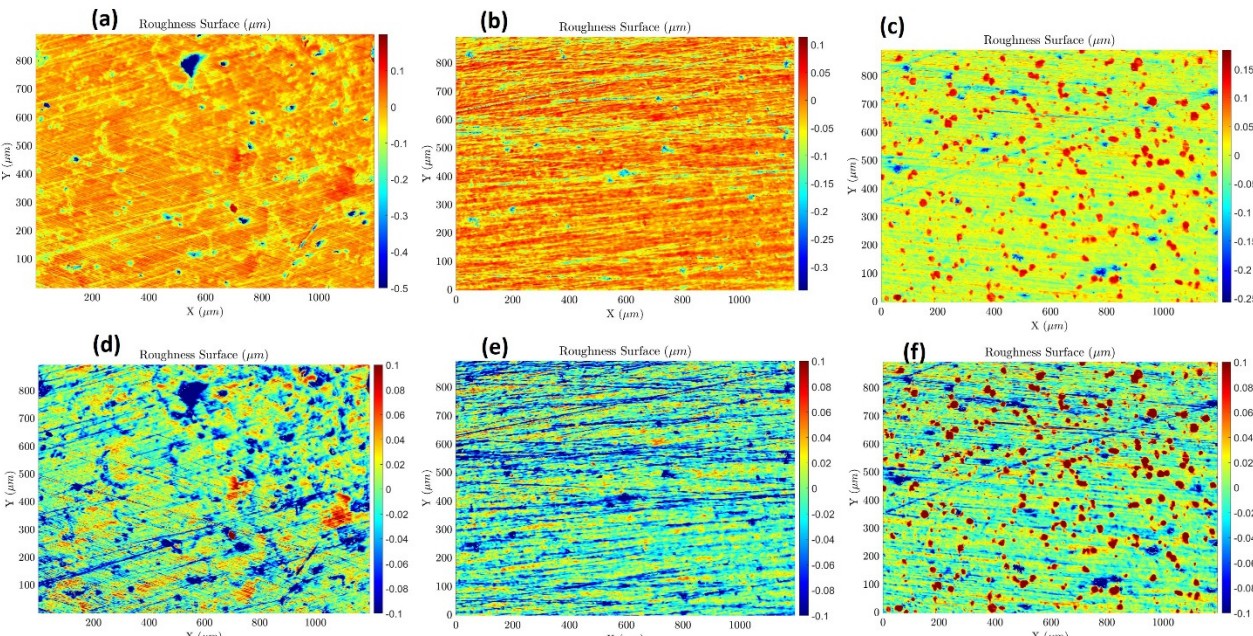

**Figure 6.** Surface elevation map obtained by WLI for the three tested coatings after surface finishing: (**a**) 13Mn6 LDS coating; (**b**) 410L cold spray coating; (**c**) 410L + 20%vol. M2 cold spray coating. (**d**–**f**) correspond to (**a**–**c**) but height-zoomed to −0.1 to +0.1 μm for easy visualization.

**Table 1.** Surface roughness and features.

| Parameter | Unit | LDS | 410L | 410L + 20% M2 |
|---|---|---|---|---|
| Sa | nm | 53 | 32 | 44 |
| Spk | nm | 64 | 30 | 102 |
| Sk | nm | 91 | 89 | 105 |
| Svk | nm | 271 | 72 | 87 |
| Pore surface ratio | % | $2.2 \pm 0.6$ | $0.7 \pm 0.4$ | $0.8 \pm 0.4$ |
| Density | $1/\text{mm}^2$ | $292 \pm 117$ | $511 \pm 193$ | $385 \pm 99$ |
| Avg. area | $\mu\text{m}^2$ | $84 \pm 34$ | $14 \pm 4$ | $20 \pm 7$ |
| Avg. volume | $\mu\text{m}^3$ | $64 \pm 73$ | $0.5 \pm 0.3$ | $1.5 \pm 1.1$ |
| Protuberance surface ratio | % | $0.79 \pm 0.95$ | | $7.5 \pm 0.5$ |
| Density | $1/\text{mm}^2$ | $1.40 \pm 1.48$ | | $208 \pm 11$ |
| Diameter | μm | $75.5 \pm 36.1$ | | $18.8 \pm 7.1$ |
| Avg. height | nm | $184 \pm 93$ | | $119 \pm 15$ |
| Max. height | nm | $338 \pm 248$ | | $198 \pm 22$ |

## 3. Results

### 3.1. Tribological Tests

In the PhD thesis of Laurent Aubanel [20], a lubricated reciprocating tribological test was performed on the three coatings. In this test, a pin was moved on a linear alternate trajectory onto a planar substrate. The reader is referred to [20] (in French) for a detailed description of the experimental setup and results. Here, we present only a brief summary of those tests. Coated and polished plates were immersed in an oil bath (0W16, fully formulated and containing ZDDP and MoDTC) at a temperature of 100 °C. The moving counterpart was a pin made of AISI 52100 machined from a bearing ball with a contact radius of 30 mm, and a normal force of 77 N was applied. Only the pin and its holder were moving on a line, with sinusoidal kinematics, and the reciprocating sliding stroke length was 10 mm. Prior to assembly, both the flat liner samples and the pins were ultrasonically cleaned and degreased in a solution of petroleum ether. The evolution of the friction coefficient with time for the composite cold sprayed coating systematically showed

different phases. After an initial unstable period, it stabilized at a value around 0.11. After a certain time, it dropped to its final value of around 0.03/0.04, which was maintained for testing over several hours. The whole test consisted of two phases: first, a 16 min cycle at 5 Hz corresponding to a mean speed of 0.1 m.s$^{-1}$; then, the frequency was modified, taking several values (15, 10, 5, 2 and 1 Hz), corresponding to average sliding speeds ranging from 0.3 to 0.02 m.s$^{-1}$. Each frequency was maintained until the friction coefficient stabilized for more than one minute. The aim of frequency variation was to change the thickness of the oil film and study the effect of the sliding speed within the mixed/boundary lubrication regimes. The Hersey number proportional to the oil film thickness was calculated as $\eta \times V/P$, where $\eta$ is the dynamic viscosity of the oil, $V$ is the average sliding speed and $P$ is the average pressure calculated using the real contact area observed on the pins. The whole test was repeated two times per sample to confirm the reproducibility of the results. Figure 7 summarizes the results obtained for the LDS and composite 410L + 20% M2 coating. Each point in the figure corresponds to the stabilized friction coefficient (i.e., the average value over one minute after it had stabilized) at different frequencies. It must be noted that even when changing the sliding speed and temperature, the system remained in the mixed/boundary lubrication regime. The 410L cold spray coating is not presented in Figure 7 because strong adhesion wear occurred after a short sliding time. The LDS coating showed a friction coefficient of around 0.15 for the lowest Hersey number tested, decreasing down to 0.10 at higher Hersey. In contrast, friction on the cold sprayed composite coating was more stable and about three times lower. The mentioned article also reports the absence of wear due to the formation of a very efficient tribofilm on the composite surface, assuring impressive performances in the mixed/boundary lubrication regime. The aim of the present work was to extend the experimental results to the hydrodynamic regime by numerical simulation in order to investigate the potential performance of such a coating under all the tribological conditions encountered in the actual system.

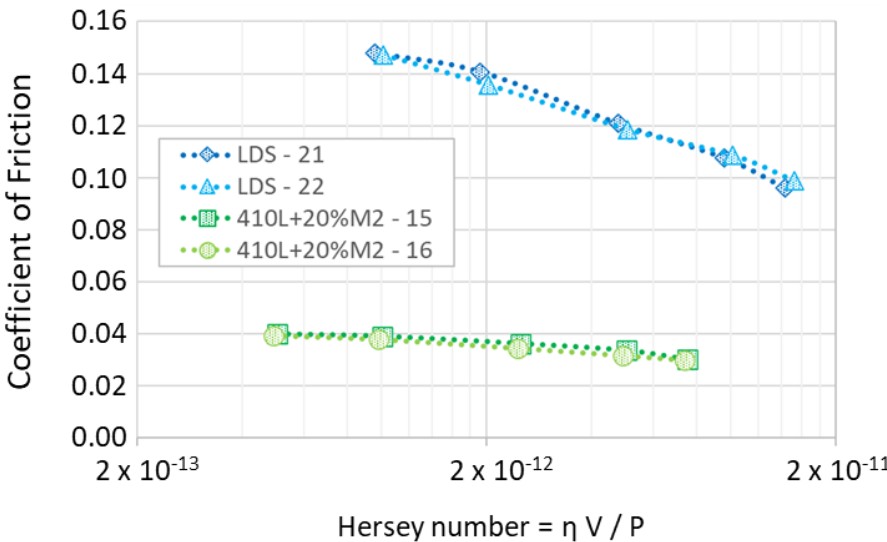

**Figure 7.** Final stable friction coefficient at different Hersey numbers for LDS and cold spray composite coatings. Results of two tests under identical conditions for each coating are reported to show the good repeatability of the measurements.

### 3.2. Surface Characterization

New measurements of the surface topographies from the tested samples were used in the simulations. The samples were measured by a white light interferometer, and data were filtered by the commercial software delivered with the equipment. Table 1 shows the average and standard deviation of 10 surface measurements, which are reported in [20]. Figure 8 shows the surface height map of two of 410L + 20% M2 cold sprayed samples.

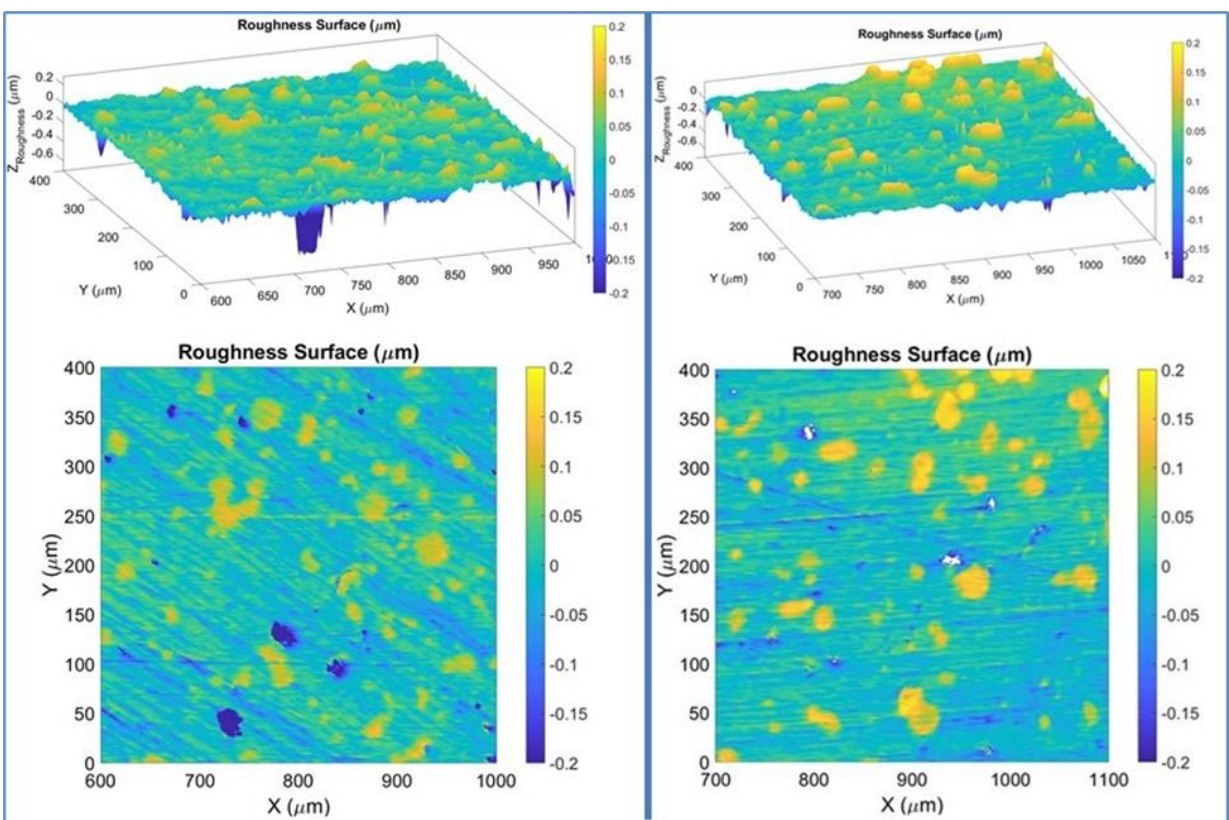

**Figure 8.** Topography measurements of two 410L + 20% M2 cold sprayed samples. The protuberances are composed of hard phases.

### 3.3. Computer Simulation

During actual engine operation, the cylinder bore surfaces encounter speeds, stroke lengths and loads that differ from those applied in the tribological test. Figure 9 shows the computational procedure used to investigate the localized effect of dimples and protuberances:

- Additional roughness filtering of surface form and waviness than can impact the contact. The measured topography is divided into 200 μm slices to minimize the CPU effort and to mimic the ring contact area;
- A deterministic model of the mixed regime in which surface separations, oil viscosity and speed are imposed to calculate the effect of the surface on oil flow, pressures and contact;
- Use of the averaged slice results, also known as "correlation factors", from the deterministic model on a reciprocating ring/bore code with speeds, loads and oil viscosity values representative of the actual system to predict the instantaneous oil film thickness, friction losses, etc.

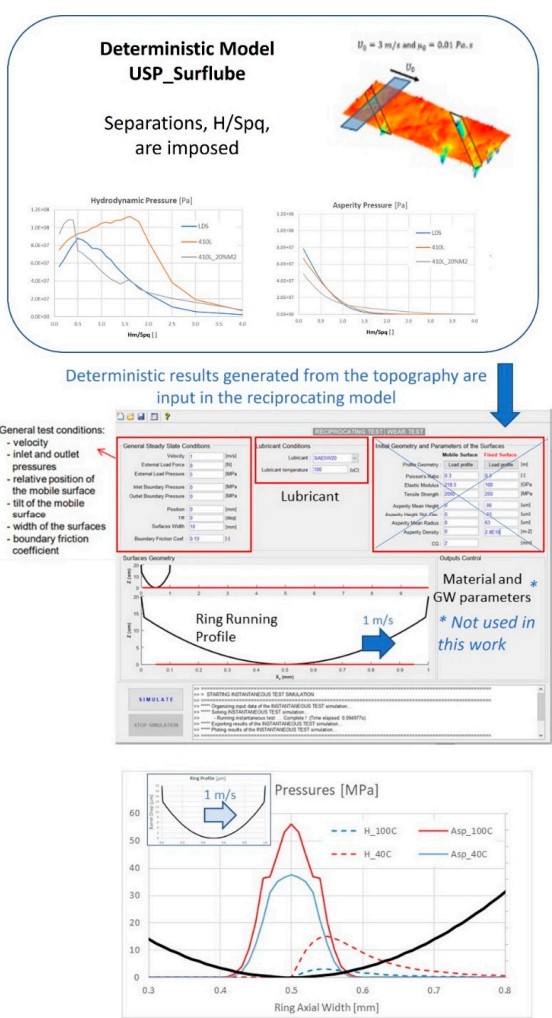

**Figure 9.** Scheme of the applied simulation. Adapted from [17].

3.3.1. Deterministic Model

The in-house deterministic Surflube model (see details in [17,21]) was used to simulate the topographies under a mixed lubricant regime. The main limitations of the model for the actual case are:

(a) Material properties are considered identical for the three coatings;

(b) The counterpart (the piston ring) is assumed to be perfectly smooth;

(c) The boundary CoF is assumed to be constant and equal to 0.10, which was later relaxed in the reciprocation simulation to consider the effect of the oil additives. Details are reported below.

Asperity Contact Modeling

The pressure that arises when an asperity is brought into contact with a smooth rigid counter surface is calculated by assuming an elastic perfectly plastic model, the pressures of which in the elastic regime are calculated using the Hertz theory for parabolically shaped bodies, with plastic deformations limited by the surface hardness of the softer material. Accordingly, the real contact area ($A_s$) and the mean contact pressure ($p_{ASP_s}$) for each region at a given surface separation are expressed as:

$$A_s = \pi \beta_s w_s \tag{1}$$

$$p_{ASP_s} = \begin{cases} \dfrac{4E_s^*}{3\pi}\left(\dfrac{w_s}{\beta_s}\right)^{1/2}, & p_{ASP_s} \le H_V \\ H_V, & p_{ASP_s} > H_V \end{cases} \tag{2}$$

where $w_s$ is the contact interference (see Figure 10), $\beta_s$ is the local asperity radius of curvature of the "region in contact", $E_s*$ the combined elastic modulus (in the current model, still considered constant along the material) and $H_V$ is the hardness of the softer material. Unlike the approach adopted in the original model [17,21], in the current model, $\beta_s$ is determined by approximating the geometry of each region in contact with a best-fit parabolical cap using the neighboring points of the highest point in contact. Furthermore, in contrast to the original model, in which the contact pressure was assigned to the highest peak of a given asperity, in the current model, the contact pressure is distributed over the points that compound the asperity following the Hertzian profile shown in Figure 10.

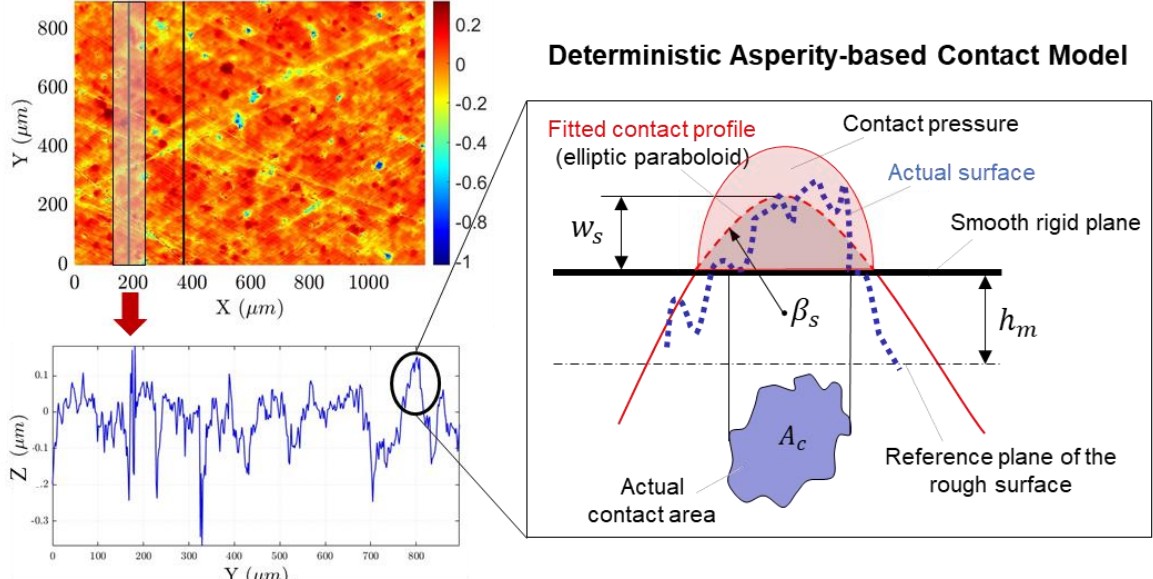

**Figure 10.** Schematic illustration of the model to define local contact interactions.

One example of the calculated $Pasp_s$ in the previous and current model is shown in Figure 11. While the previous Greenwood-based model showed only the discrete asperity contacts, the updated model used in this work was able to better reproduce the larger contact caused by the protuberances.

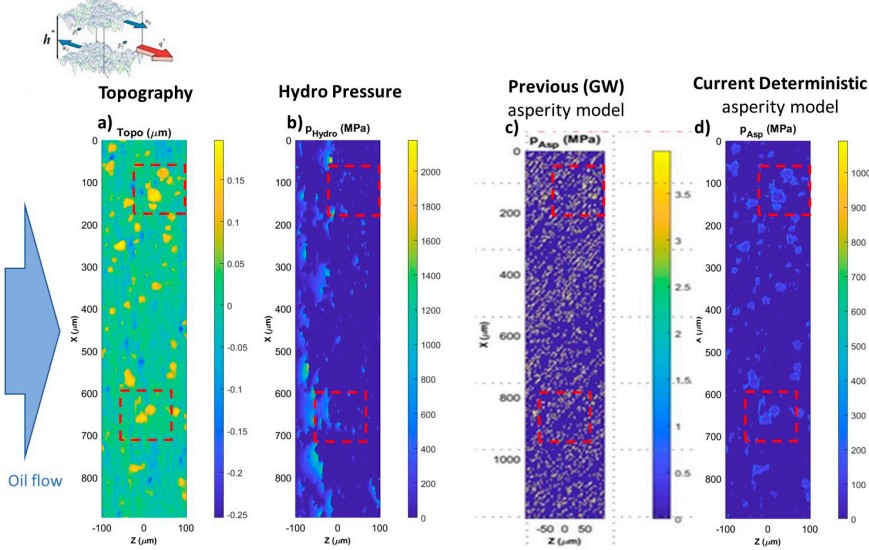

**Figure 11.** Simulation results: (**a**) topography; (**b**) hydrodynamic pressure; (**c**) asperity pressure according to the GW model; (**d**) asperity pressure according to the deterministic asperity-based model.

Hydrodynamic Friction Modeling

As mentioned before, the hydrodynamic regime was modeled by solving the Reynolds equation of each surface node with the following imposed values:

- Oil viscosity: 0.01 Pa·s;
- Oil flow speed: 3 m/s;
- Surface separations: 0.1 to 10× Spq.

For the current work, no shear thinning or oil critical shear rate was adopted. Further studies are planned for when more experimental data are available.

Figure 12 illustrates the deterministic simulation results for the three coating surfaces. The figures show (from top to bottom): the measured topography, the hydrodynamic and asperity pressures calculated for a surface separation of approx. 0.05 μm with the imposed oil flow and viscosity used in the simulation. Regions with higher Pasp (light blue to yellow colors) appear, as expected, in the regions with higher height, while higher hydrodynamic pressure occurs "in front" of these regions due to the convergent profile. As a result, for these analyzed surface slices and separations, the very smooth 410L presents relatively high hydrodynamic support and low asperity pressure. Nevertheless, such hydrodynamic pressure combined with very low surface separation also causes a very high shear rate. Figure 13 shows the average hydrodynamic and asperity pressures for the surface "slices" reported in Figure 12. Figures 14 and 15 show the average pressures for the different surface separations in terms of the oil film parameter, i.e., t = Hm/Spq, where Hm is the oil film thickness, and Spq is the plateau root mean square roughness.

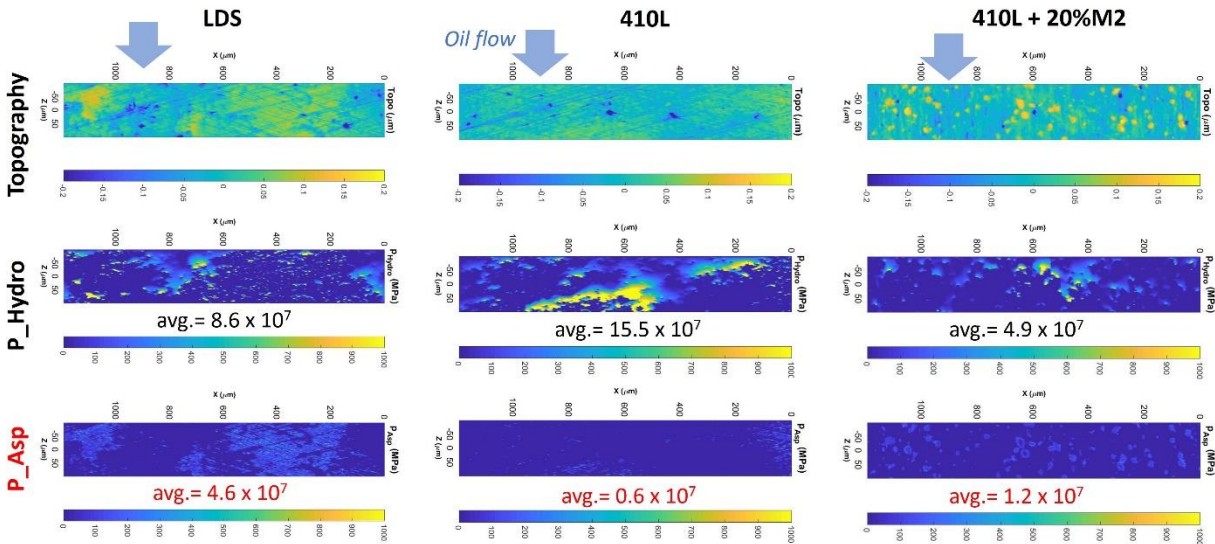

**Figure 12.** Example of the deterministic simulation results (@ Hm ~ 0.05 μm).

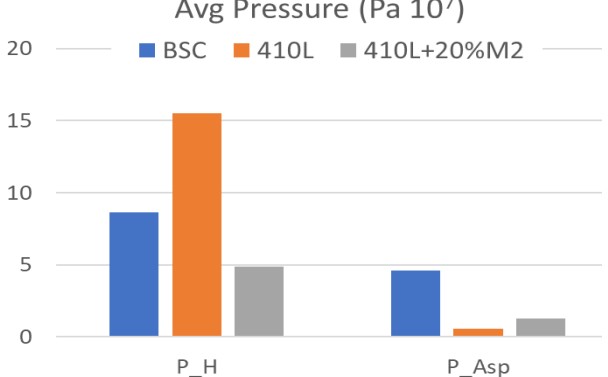

**Figure 13.** Hydrodynamic (P_H) and asperity(P_Asp) pressure results (@ Hm ~ 0.05 μm).

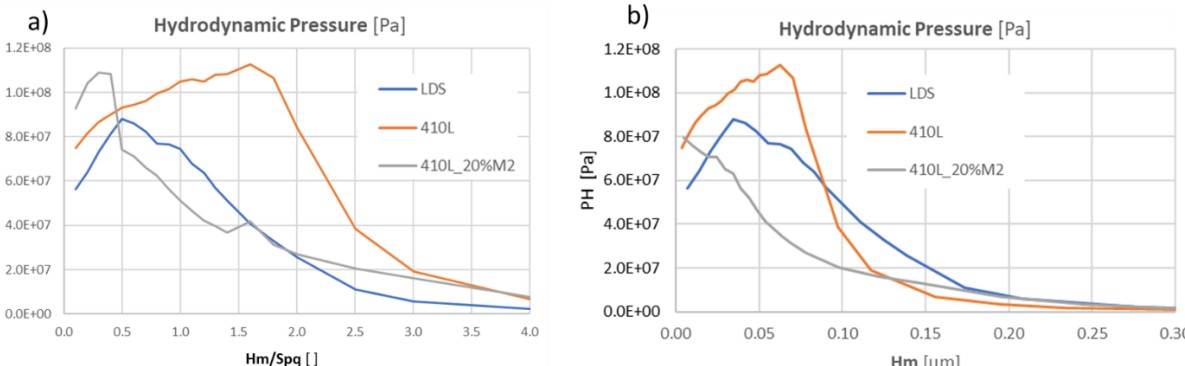

**Figure 14.** Average hydrodynamic pressure according to the deterministic simulation. (**a**) Oil film results normalized by the respective Spq; (**b**) same data, but the X axis represents oil film thickness.

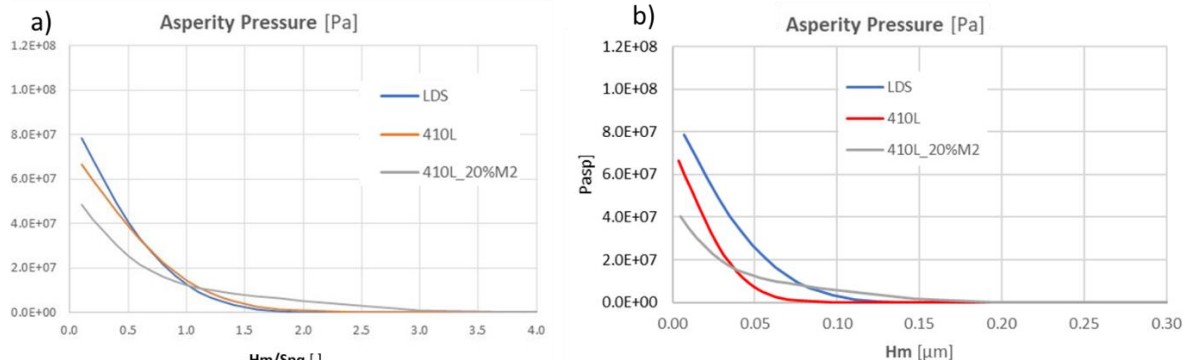

**Figure 15.** Average asperity pressure according to the deterministic simulation. (**a**) Oil film results normalized by the respective Spq; (**b**) same data, but the X axis represents oil film thickness.

### 3.4. Piston Ring Reciprocating Simulation

In the previously described deterministic model, surface separations and oil flow speed were imposed (such as the Patir & Chang model). To investigate the more complex and transient regimes found in the engine piston ring, the in-house simulation code VTL was used. Figure 16 shows a screenshot of the user interface of the VTL software. A 0.2 mm wide flat profile (emulating the oil control ring land) at different working conditions was used as input. In this approach, the correlation factors calculated by the deterministic model, instead of the surface statistical parameters, were provided as input to the VTL reciprocating module.

The simulation conditions (load and speeds) were arbitrarily defined to create "Stribeck-like" curves covering the three main lubrication regimes (boundary, mixed and hydrodynamic). The oil viscosity of SAE $0W-16$ at 90 °C was imposed. The ring was assumed to be flat and smooth.

For a more realistic comparison with production engine blocks, surface deterministic correlation factors from an actual mirror-like LDS engine block, instead of the polished LDS previously presented, were used in the reciprocating simulations. Data and details of the LDS engine block can be found in [17].

For the composite cold spray, two cases were simulated:

- M2 (CoF 0.10): data from the 410L + 20%, using the boundary CoF = 0.10 of the 410 and the LDS cases as input;
- M2 (CoF 0.04): same as before, with the asperity friction correlation factors multiplied by 0.40 to reproduce the experimental values when using oil FM, as presented in Figure 7.

Figure 17 shows the average CoF for each surface.

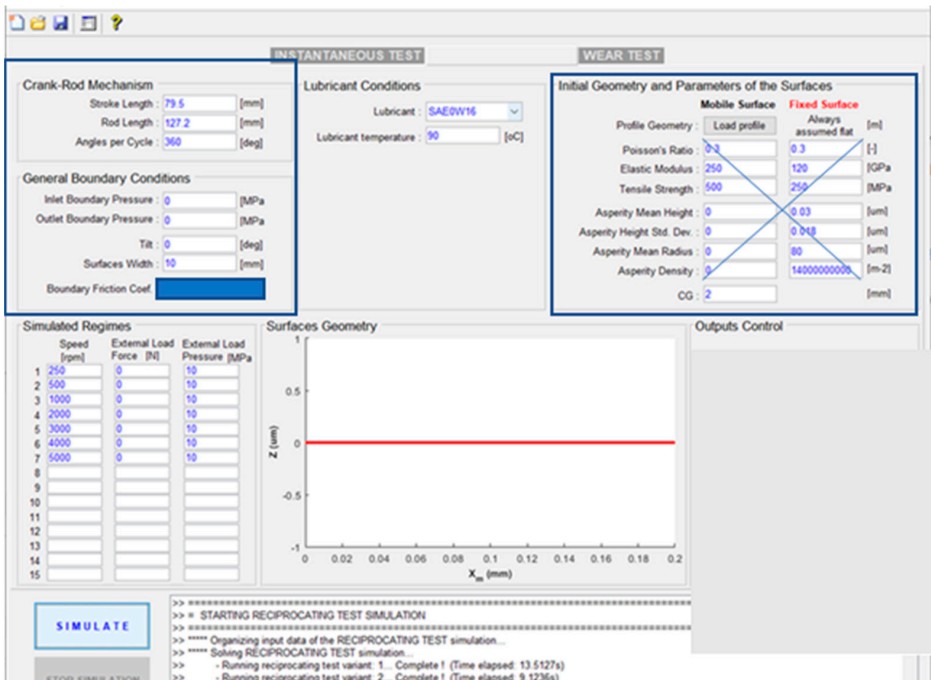

**Figure 16.** Reciprocating model (VTL) interface and applied conditions.

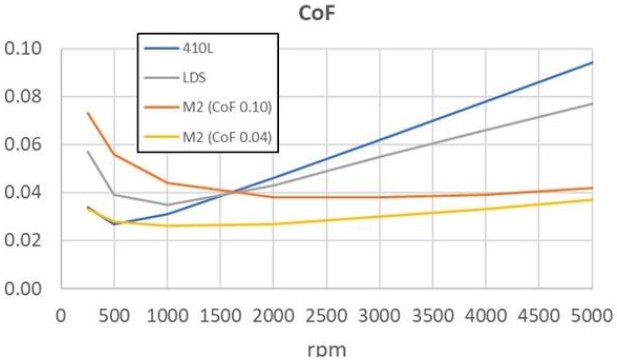

**Figure 17.** Coefficient of friction of the different surfaces; 410L and LDS were simulated with CoF = 0.10.

## 4. Discussion

Upstanding harder material (e.g., burrs) in iron-based mirror-like bore spray coating is a feature controlled by material compositions that can be studied thanks to the cold spray process. The protuberances (working as microplateaus) seem to have similarities with silicon crystals or carbides in aluminum alloys or plated versions thereof. Beyond the difference in the base material, the hardness value level and its gradation in the material composition also differ, as well as the additional feature of evenly distributed pores. This combines different phenomena of sliding friction on behalf of lubrification, as well as aspheric pressure. Tribofilm generation on the summits can be expected to be more effective than in conventional cylinder bore variants, which encourages further research and optimized deterministic simulation possibilities for that purpose.

The current investigation showed that the cold spray with protuberances, although rougher, can minimize friction in all engine regimes as follows:

- In boundary/mixed lubricant regimes, the protuberances help to create a MoS$_2$ tribofilm when using friction modifiers (FMs) in the oils. The results and mechanisms were validated by experiments [20];
- In the mixed/hydro regime, where the friction losses are dominated by the oil shear, the protuberances help to create a thicker oil film, reducing the shear rate, as illustrate in Figure 18.

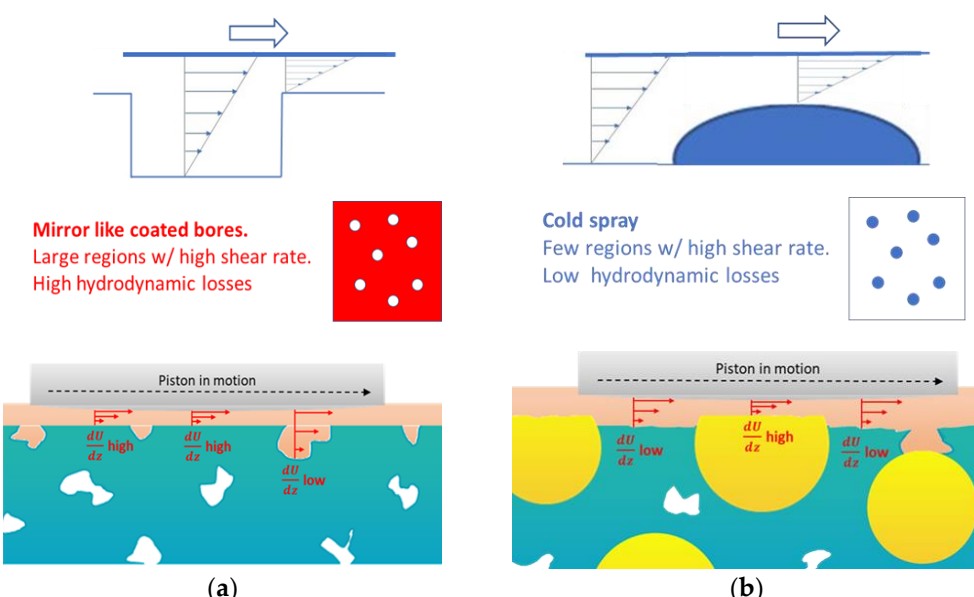

**Figure 18.** Schematic view of the local variations of the oil shear rate for two surfaces: (**a**) with dimples (as in the BSC coating) and (**b**) with protuberances (as in the 410L + 20% M2 coating).

Figure 19 shows instantaneous CoF and oil film thickness, as well as the hydrodynamic and asperity friction losses in the reciprocating simulation at 1000 rpm. For simplicity, the load and temperature assumed in the simulation for downward and upward strokes were identical. Under actual engine conditions, there would be some differences, but the main conclusions of the simulation remain valid.

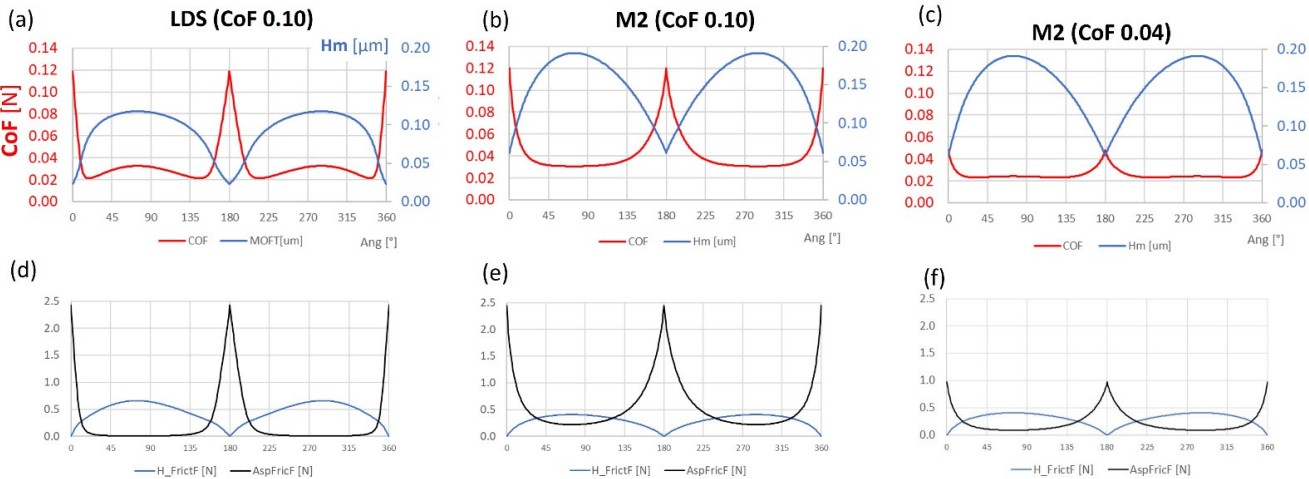

**Figure 19.** Simulation result at 1000 rpm. Maximum slide speed = 3.4 m/s. (**a**–**c**) Instantaneous CoF and oil film thickness along the stroke; (**d**–**f**) hydrodynamic and friction forces along the stroke.

For LDS, because its surface is smoother, its oil film is thinner, and asperity forces are relevant only close to the reversal points.

In M2 (CoF 0.10), because it is "rougher", its oil film thickness and asperity contact are higher than those of LDS.

For M2 (CoF 0.04), when using the CoF value measured in [18,19] with FM oil additives, oil film obviously remains identical, but friction is significantly reduced due to the lower asperity friction forces.

Figure 20 shows the same plots at 5000 rpm. For all cases, the oil film thickness is reduced, especially at mid-stroke, due to be more efficient hydrodynamic support on the flat counterpart (On a profiled counterpart, the situation is different; oil film increases with

speed). The oil film thinning is more evident in the smoother LDS, since for the other surfaces, a minimum oil film thickness is guaranteed by the height of protuberances.

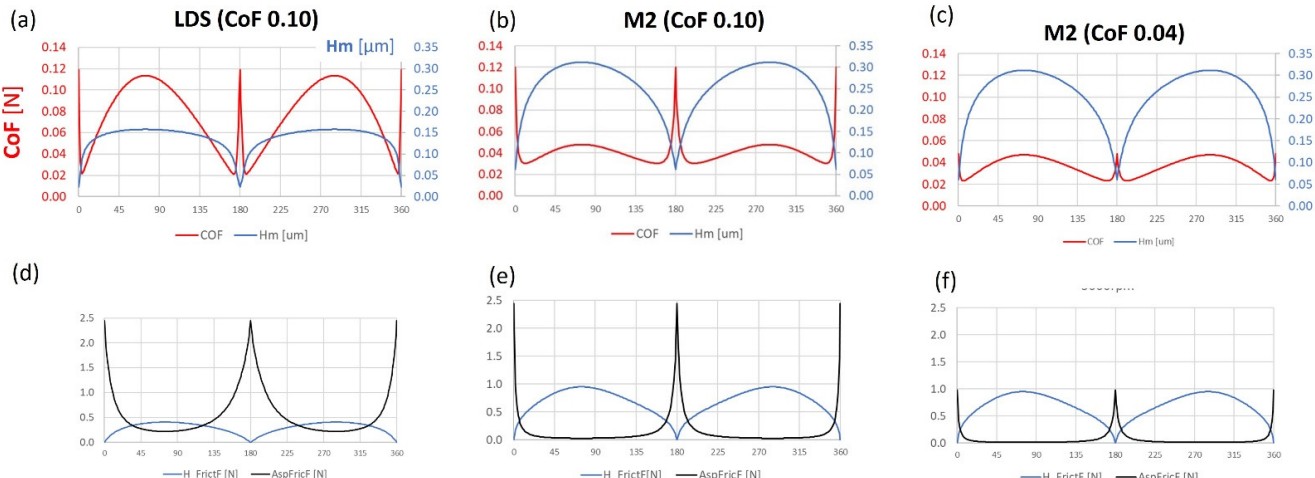

**Figure 20.** Results at 5000 rpm. Maximum slide speed = 21.8 m/s. (**a**–**c**) Instantaneous CoF and oil film thickness along the stroke; (**d**–**f**) hydrodynamic and friction forces along the stroke.

Due to the higher shear rate of LDS, hydrodynamic friction increases significantly at mid-stroke (i.e., higher slide speeds). Asperity contact only occurs very close to the reversal points.

For M2 (CoF 0.10), the increase in the oil film is enough to significantly reduce the asperity contact. Most of the load is now supported by hydrodynamic pressure instead of the asperity pressures. At 5000 rpm, the instantaneous CoF is lower than that of the LDS, except in very localized zones close to the reversal points.

For M2 (CoF 0.04), the oil film thickness and the hydrodynamic friction are identical to the M2 case, with a CoF of 0.10. The reduced CoF = 0.04 obviously reduces the instantaneous asperity friction. Nevertheless, as the average friction is now dominated by the hydrodynamic share, only a small reduction in the cycle averaged CoF is observed.

Figure 21 extends the simulated CoF to 10,000 rpm. At higher speeds, the friction with smoother surfaces increases significantly due to the thinner oil film, resulting in a higher shear rate. On the other hand, the surfaces with protuberances, again assuring a thicker oil film, show only a moderate increase in friction with speed. At higher speeds, the friction is dominated by hydrodynamics, and the effect of the oil FM in reducing the asperity friction becomes less influential.

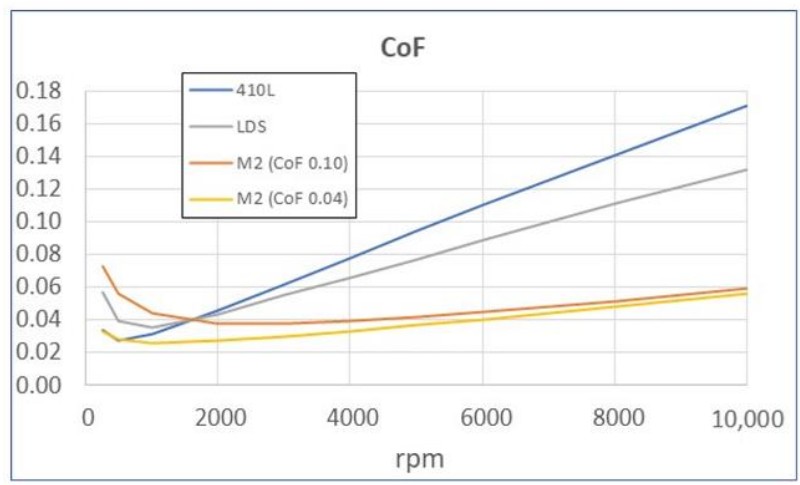

**Figure 21.** CoF results up to 10,000 rpm.

## 5. Conclusions

According to the simulation results, the cold spray composite coating had the lowest friction at higher speeds due to its low hydrodynamic pressure and oil shear rates.

At low speeds, when friction is dominated by asperity contact, the composite cold spray has the highest friction if tested without FM oil additives.

The high friction at low speeds can be mitigated by reducing the boundary CoF, e.g., with the use of FM oil additives, as demonstrated in the tribological tests.

These conclusions can be applied to other surfaces with protuberances with "system-optimized" dimensions.

At higher speeds the effect of the FM is reduced and becomes negligible.

**Author Contributions:** Conceptualization, E.T.; methodology, L.A. and E.T.; investigation, L.A., E.T. and M.B.; writing—original draft preparation, E.T.; writing—review and editing, E.T., L.A., M.B., F.D. and T.L.; supervision, F.D. and T.L. All authors have read and agreed to the published version of the manuscript.

**Funding:** This research received no external funding.

**Data Availability Statement:** Proprietary data was used.

**Acknowledgments:** The authors thank Francisco Profito for his always expeditious support and enlightening discussion about the computer simulations.

**Conflicts of Interest:** The authors declare no conflict of interest.

## Appendix A. Surface Characterization and Function Orientated Analysis

The investigation carried in this work reinforces the need for improved surface characterization of localized features such as burrs and protuberances. Figure A1 illustrates the common statistical approach, as shown in Table 1.

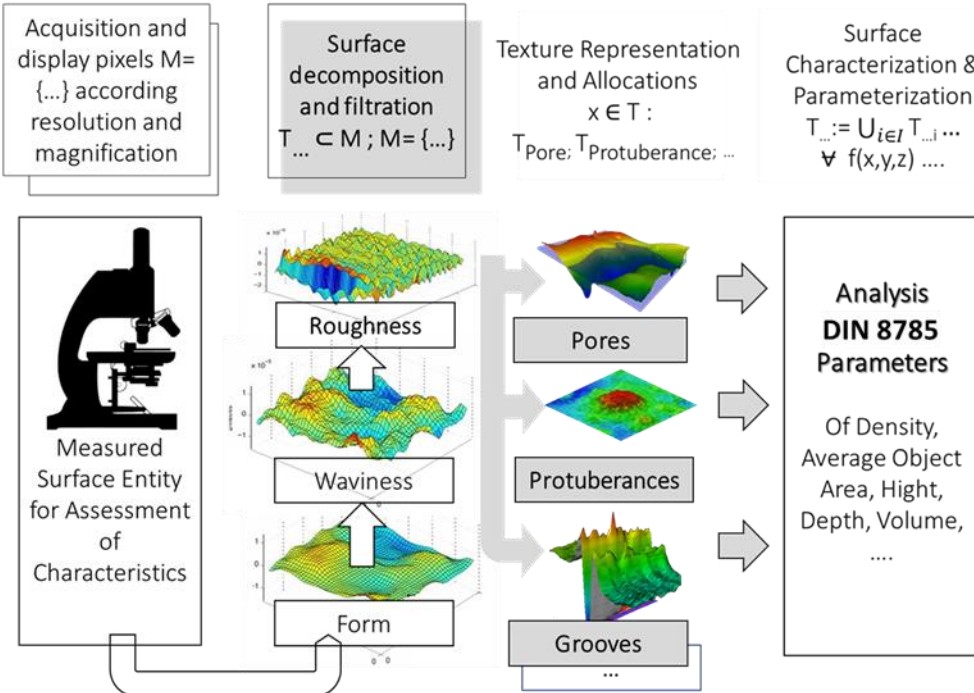

**Figure A1.** Scheme of detailed surface characterization.

However, statistical parameters may fail to identify localized effects (see Figures 10–12). More resolute and robust methodologies are needed to identify and quantify localized

features (or objects) such as pores, protuberances or other "surface textures". During the development described in the current work, more detailed surface analysis was used to optimize the coating and surface finish of the prototypes, as well for honed tubes, which will be discussed in a future paper.

A description of the more detailed surface characterization is outside the scope of this paper and can be found in [14] and in an author forthcoming PhD thesis. Figure A2 illustrates the methodology, and Figure A3 shows some of the results.

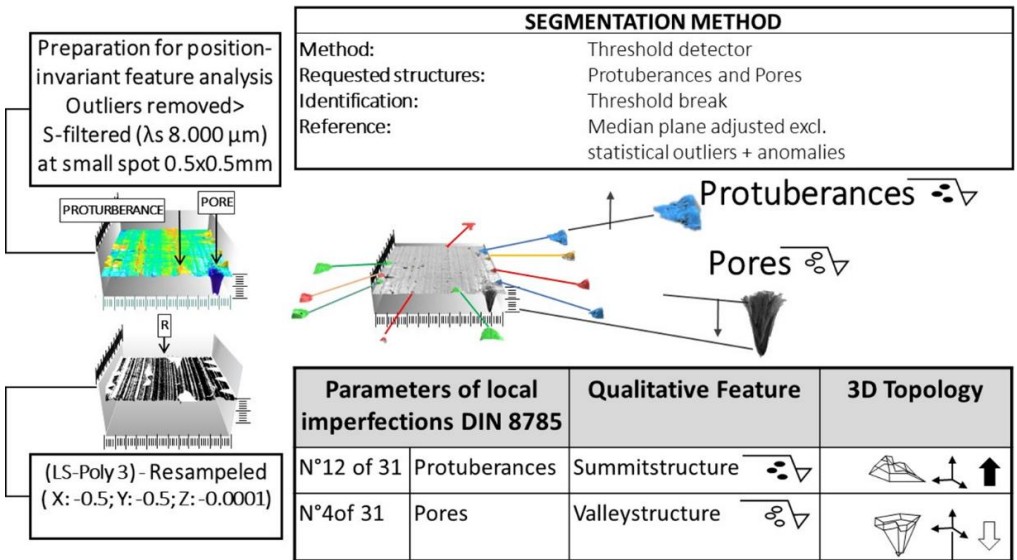

**Figure A2.** Scheme of the segmentation method to evidence the protuberances and pore features.

A white light interferometer (DIN EN ISO 25178-604) with an optical zoom of 50× and an X/Y/Z resolution of $0.19/0.19/10^{-6}$ µm was used. Single frames of 300 × 300 µm were stitched to a total of 522 frames with an overlap of 10% to describe the measurement area of 12 × 3 mm. Conditions for measurement and evaluation were set to expected orthogonal wavelengths in accordance with DIN EN ISO 21920-2:12/2022 and DIN EN ISO 25178-6.

The surface point cloud was resampled to avoid unwanted filter effects. Typical disturbances in the information that can be physically assigned to the measurement process were successfully reduced by resampling, followed by the segmentation of the structural features of protruding or recessed particulate delimitable objects (called "particles" in MountainsMap®) to be found locally using the method of threshold value determination over a section level of the Gaussian surface. The particles are distributed in the matrix in a position-invariant manner. A matrix search or watershed recognition using a morphological reconstruction would not lead to success here. The confidence level of the subsequent feature analysis would not reach the intended range. The recorded surface was filtered with a transmission filter according to DIN EN ISO 16610-21. The particles pass through the filter. Short-wave structural components that do not belong to particles are largely suppressed by the filter. The correct parameterization is determined by analytical methods for error reduction. Structure separation enables separate quantification of the particles to be found. The segmentation and feature analysis tools used here are in accordance with the International Geometric Organization for defining Product Specifications (GPS).

Then (see scheme on Figure A3):

1. Topography segmentation for feature extraction is applied (in this case protuberances), and the features are quantified. As an example, Figure A3b shows the protuberance density. As periodical features need to be excluded from standard roughness parameter calculation, the periodical portion of the surface can also be quantified as Figure A3c (which helps to validate each 200 µm slice of the remaining area without features);

2. A uniform Cartesian coordinate system is required to crosscheck performance parameters, e.g., the asperity contact pressure in Figure A3d, with statistics of the surface features (Figure A3b via simple mathematical translation;

3. A joint feature plot is generated (Figure A3e, which contributed to better analysis of the objects for comparison with computer simulation of contact;

4. The agreement between simulated data and theoretical knowledge/assumption can be evaluated. Figure A3d,f show asperity contact for two different simulation models. Figure A3e,g show that the surface segmentation can be used to validate the models. It is unlikely that the asperity contact does not match the segmented upstanding microplateaus. The predictions made with the new simulation model (Figure 10) are more realistic than those made with the previous model.

With this improved characterization, surface features can be more precisely valued and described for surface analysis and used as input for computer simulation. This helps to define controllable quantities for manufacturing. Areal density of segmented objects and their distributions of attributes (height or depth, widespread expansions and shape) values and limits are necessary to produce parts with the desired foreseeable performance. Such object segmentation also allows for the generation of virtual surfaces by editing the feature matrix.

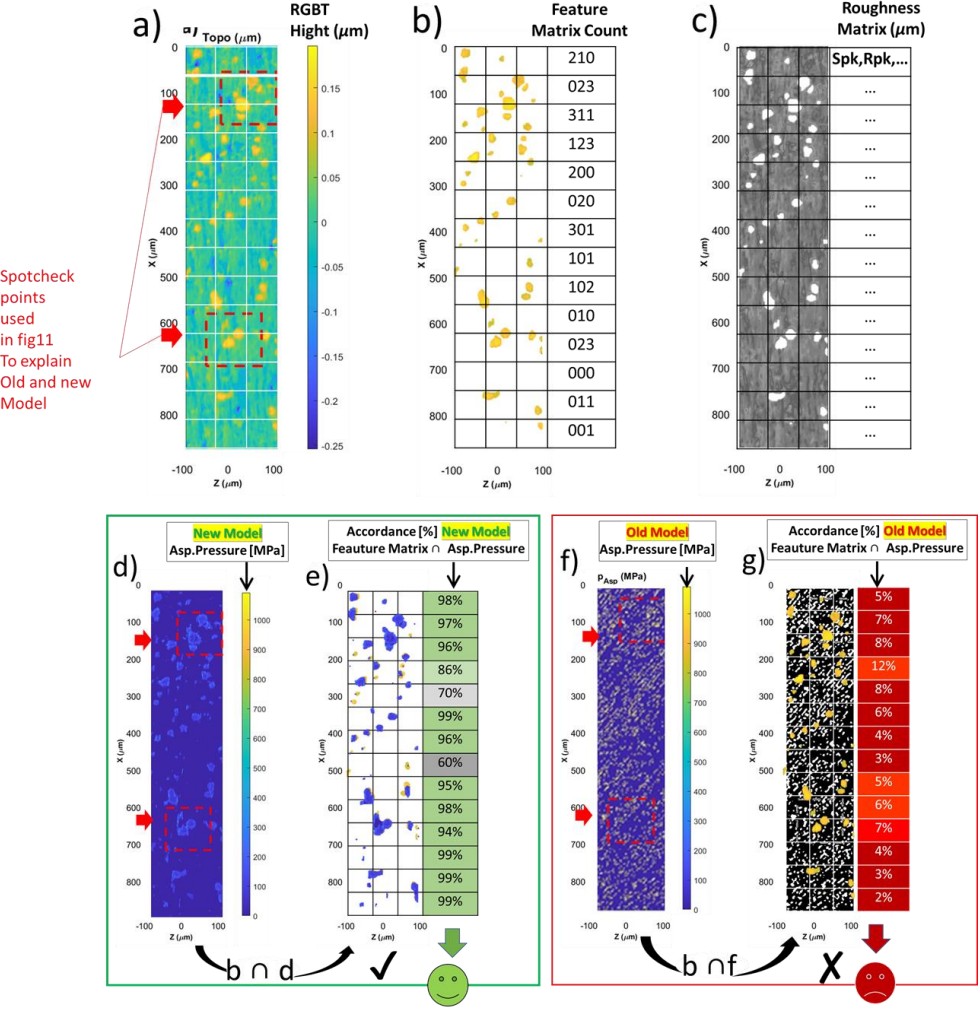

**Figure A3.** Scheme of the segmentation method to evidence protuberances and pore features. (**a**) Measured topography after form and roughness filtering; (**b**) matrix of the protuberances; (**c**) background roughness data excl. segmented objects; (**d**) contact predicted by the new asperity model; (**e**) comparison of surface and contact pressure of the new model; (**f**) contact predicted by the previous model; (**g**) comparison of surface and contact pressure of the previous model.

Using commercial codes such as MountainsMap®, one can easily obtain topography study tables for statistics (average values, standard deviation, etc., of each attribute). In order to compare these MountainsMap® data to the computed functional performance plots in a uniform Cartesian coordinate system via simple mathematical translation, an export–import handshake code is required. Then, geometric surface appearance analysis can be crosschecked with performance analysis. Explanations of the geometric surface analysis is available from Cognard on the webpage of Digitalsurf (https://www.digitalsurf.com/news/perform-a-particle-analysis-on-microscopy-images/ (accessed on 15 January 2023)). To use this commercial procedure appropriately, prior to the analysis of protuberances and pores, leveling of the surface is also necessary. To obtain interpretable results, the surface roughness must be lower than the height of the quantified objects (lower than the height of the pores and the protuberances without taking into account the +/− sign). This condition helps to ensure that pores and protuberances stand out clearly on a flat background using common thresholds. A shift in the slope of the distribution curve of height, together with lateral topological connection, combines for the correct object recognition. Contrast can be optimized by applying multiscale analyses DIN EN ISO 16610-49 and -89 and morphological filters -41 and -81.

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
