# Peer review of "Tribological Performance of a Composite Cold Spray for Coated Bores"

_lubricants, doi:10.3390/lubricants11030127_

Round 1

Reviewer 1 Report

The work presented for review concerns the study of coatings applied to the cylinder surfaces of internal combustion engines. The idea is interesting, and the manuscript itself is written in a clear and transparent way. You can see that the authors put a lot of work and time into it. However, I have some suggestions that I believe will improve the quality of the manuscript:

1. There is no information about the coating spraying process, what machine, what parameters, etc. Please add it
2. There is also no information on how coat was polish with sandpaper, was a machine used? Or was polishing done by hand? What was the polishing time, what pressure? Is a gradation of 1200 enough? With lower grits, of the order of 2500 or 3000, the coating would be smoother. Have the authors considered using finer paper?
3. How were SEM images of the coating obtained? Has the cylinder been cut open? Or is it a coating applied to some samples? There is not sufficient explenation.
4. The authors write that the pin was used in tribological studies. However, there is no mention of the test stand, was it a pin-on-disc stand? or some other method?

5.Why  77N was chosen as the normal force?
6. What were the other parameters of the experiment, i.e. contact pressure, friction path, pin linear speed?
7. Fig. 7 shows the values of tcof. What was the measurement scheme, i.e. is it an average value? Has the run-in period been included in this value?
8. The name of Figure 9 is placed on line 260, please move it under the figure, one page earlier
9. Section 3.3.1.2 provides test assumptions, such as oil viscosity. Why were these values chosen? Has the fact that these values change (particularly viscosity) with temperature been taken into account?

Author Response

We thank the reviewers for their constructive remarks. We would like to explain them that many details on the coating manufacturing and the tribological experiments were not included in the present article because the authors submitted another article/journal. We hoped to use that reference recurrently in the present article, but this is not possible today. So, we decided to follow the suggestions of reviewer 1 and we added more details concerning the coating manufacturing and the tribological tests.

Reviewer 1

  1. There is no information about the coating spraying process, what machine, what parameters, etc. Please add it

Several details on the process and materials were added in page 4 of the revised manuscript.

The two commercially available steel powders, produced by Sandvik Osprey Ltd, were mixed with a respective volume ratio of 80/20. Their particle size distribution, as given by the supplier, was +32-10 µm. and sprayed to form the composite coating directlyThe cold spray equipment used was a 5/11 from Impact Innovations GmbH, equipped with an internal diameter coating device and using nitrogen as principal gas, with a pressure of 5 MPa and a temperature of 900 °C. The stand-off distance and powder feeder rotation speed were respectively set at 5 mm and 8 rpm. They were sprayed to form the composite coating directly. onto 10-mm-thick laminated aluminum 6060 plates with dimensions of 65 by 40 mm.

  1. There is also no information on how coat was polish with sandpaper, was a machine used? Or was polishing done by hand? What was the polishing time, what pressure? Is a gradation of 1200 enough? With lower grits, of the order of 2500 or 3000, the coating would be smoother. Have the authors considered using finer paper?

An explanation was added in page 7 of the revised manuscript. The text now reads as follows:

The surface finish for the three coatings was made by polishing with 1200 SiC abrasive paper, using a standard metallographic preparation polish machine with water as lubricant and cleaning fluid. After different trials, this procedure was chosen because it resulted in surface textures close to those obtained by honing, which is the process traditionally used in engine industrial production.

  1. How were SEM images of the coating obtained? Has the cylinder been cut open? Or is it a coating applied to some samples? There is not sufficient explenation.

Please, see our answer to question n. 1. Among the details added, we specified that substrates were plates and not the real cylinder bore part. Indeed, the cold spray “internal diameter” nozzle was used to produce those coatings on planar samples, as described in the revised text. The same configuration (cold spray machine, nozzle, parameters and powders) can thus be used for coating the real cylinder bore part.

  1. The authors write that the pin was used in tribological studies. However, there is no mention of the test stand, was it a pin-on-disc stand? or some other method?

The test stand was a lubricated reciprocating  machine (linear alternate movement of a pin on a plate). The tribological test set-up and results are fully described in the article we submitted many months ago (as explained before) and not yet published. The same contents are available in L. Aubanel’s PhD thesis, which is in the public domain and open access. Our concern here is to keep that information to a minimum, because it concerns the mixed and boundary regime, while the aim of the present article is to give insights into the hydrodynamic tribological regime. The text was modified as follows at page 8 of the revised manuscript

In the PhD thesis of Laurent Aubanel [19], a lubricated reciprocating tribological test was performed on the three coatings. In this test, a pin is moved on a linear alternate trajectory onto a planar substrate. The reader is referred to [19] (in French) for a detailed description of the experimental set-up and results. Here, we are presenting only a brief summary of those.

5.Why  77N was chosen as the normal force?

The test was developed by a team of tribologues in Renault prior to the authors’ work. Its aim was to screen more easily coatings and topographies compared to an already well know ring-on-liner test. The different parameters of the tribological test (including the normal force) were studied on reference samples. The defined parameters allowed to obtain comparable results compared to the (more complex) ring-on-liner test. In the present work, we applied the parameters developed previously (internal study in Renault and not published). We are sorry but being a private company information, we would prefer not to communicate this information in the paper and simply leave the value as it is.

  1. What were the other parameters of the experiment, i.e. contact pressure, friction path, pin linear speed?

The tribological test has been described with more details in the revised manuscript (page 8 and 9), as follows:

Coated and polished plates were immersed into an oil bath (0W16, fully formulated containing in particular ZDDP and MoDTC) at a temperature of 100 °C. The moving counterpart was a pin made of AISI 52100, machined from a bearing ball with a contact radius of 30 mm, and a normal force of 77 N was applied. Only the pin and its holder were moving on a line, with sinusoidal kinematics, and the reciprocating sliding stroke length was 10 mm. Prior to assembly, both the flat liner samples and the pins were ultrasonically cleaned and degreased in a solution of petroleum ether. The evolution of the friction coefficient with time for the composite cold sprayed coating systematically showed different phases. After an initial unstable period, it stabilized at a value around 0.11. After a certain time, it dropped to its final value around 0.03 / 0.04, which was maintained even for testing over several hours. The whole test consisted of two phases: first, a 16 min long cycle at 5 Hz, corresponding to a mean speed of 0.1 m.s-1; then, the frequency was modified, taking several values (15, 10, 5, 2 and 1 Hz), corresponding to average sliding speeds ranging from 0.3 to 0.02 m.s-1. Each frequency was maintained until the friction coefficient stabilized over one minute. Frequency variation aimed at changing the thickness of the oil film and at studying the effect of the sliding speed within the mixed/boundary lubrication regimes. The Hersey number, proportional to the oil film thickness, was calculated as η * V / P, where η is the dynamic viscosity of the oil, V the average sliding speed and P the average pressure, calculated using the real contact area observed on the pins. The whole test was repeated two times per sample, to confirm the reproducibility of the results.

  1. Fig. 7 shows the values of tcof. What was the measurement scheme, i.e. is it an average value? Has the run-in period been included in this value?

With the more detailed presentation of the tribological test (see previous point), Fig. 7 should now be clearer to the reader. The friction coefficient considered is only the one after the final stabilization, so it does not include the run-in period (neither the first “pseudo-stable” phase). The following explanation was added to the revised document (page 9):

Figure 7 summarizes the results obtained for the LDS and composite coating 410L+20%M2. Each point in the figure corresponds to the stabilized friction coefficient (i.e. the average value of it over one minute, after it had stabilized) at different frequencies. It must be noted that even when changing sliding speed and temperature, the system remained in the mixed/boundary lubrication regime.

  1. The name of Figure 9 is placed on line 260, please move it under the figure, one page earlier

Final layout comes only after editorial work. Current version as the legend below the figure in the same page, but we can not control that,

  1. Section 3.3.1.2 provides test assumptions, such as oil viscosity. Why were these values chosen? Has the fact that these values change (particularly viscosity) with temperature been taken into account?

SAE 0W-16 and 90C were chosen as they are typical values on current modern, fuel saving oriented engines. The change of viscosity with the temperature was taken in account in the simulation. For the 0W16 oil, viscosity at 90C was 0.0075 Pa.s (versus e.g., 0.028 at 40C)

Reviewer 2 Report

The research makes a very good impression. A full range of studies has been carried out, the experimental part and modeling correlate with each other. The simplifications used in the modeling are justified and generally accepted. The paper is in scope of Lubricants and is recommended for publication. The following are comments that may be helpful to the authors.

1. The Materials and Methods section looks incomplete. It is necessary to include the methodology of tribological tests and modeling methods. Or, in order not to change the structure of the text, you can rename section 2.

2. At the end of section 1, it is necessary to clearly define the aim of the study. There you can also describe the novelty. In the text, phrases relating to the novelty of research are found in different sections.

3. The problem of plastic deformation of asperities is solved. This plastic deformation is irreversible; the surface geometry changes; at the next contact, the conditions will be not the same as initial (up to a purely elastic contact). It would be interesting to compare the resulting surface geometry (from simulation) with the initial one, as well as to correlate the roughness parameters from Table 1 with the same parameters after the tests (after removal of the tribofilm). If it is difficult to include such data, it would be good to touch on this topic in the Discussion.

Author Response

  1. The Materials and Methods section looks incomplete. It is necessary to include the methodology of tribological tests and modeling methods. Or, in order not to change the structure of the text, you can rename section 2.

See answers and paper revisions described to Reviewer 1.

  1. At the end of section 1, it is necessary to clearly define the aim of the study. There you can also describe the novelty. In the text, phrases relating to the novelty of research are found in different sections.

Following text was added:

The current work investigates the mixed and hydrodynamic regimes of different coated surfaces, mirror like, used for engine cylinder liners. The novel surface is one obtained by composite cold spray, where the powder hard particles produced localized protuberances after surface finish. Such protuberances helped to increase the oil film, reducing friction losses at higher speeds. An improved contact model to approximate the regions in contact areas used in a deterministic model in order to better represent the relatively large contact in the protuberances, The calculated deterministic results were then used in a reciprocating simulation to mimic the behavior of a piston ring. When combined with empirical friction coefficient values obtained with fully formulated oils containing friction modifiers, the novel composite cold spray surface obtained the lowest friction for both boundary and hydrodynamic lubricant regimes.

  1. The problem of plastic deformation of asperities is solved. This plastic deformation is irreversible; the surface geometry changes; at the next contact, the conditions will be not the same as initial (up to a purely elastic contact). It would be interesting to compare the resulting surface geometry (from simulation) with the initial one, as well as to correlate the roughness parameters from Table 1 with the same parameters after the tests (after removal of the tribofilm). If it is difficult to include such data, it would be good to touch on this topic in the Discussion.

Thanks for the very pertinent comments and suggestions. Study of the surface after test was previously investigated in another author work: Profito et al. Effect of cylinder liner wear on the mixed lubrication regime of TLOCR. Tribology International 93 (2016) 723–732

We plan to return to the subject, with the improved models, when topography samples of honed (not polished) cylinder liners after tests become available. Tests are in progress and hopefully will be subject of a future paper.

In addition to the reviewers’ requests, a few typos were also corrected:

DIN 4287 and 4288 are currently included in the DIN EN ISO 21920-2:12/2022. Text was revised to consider that.

DIN EN ISO 8785 was erroneous typed as DIN 8587, correct on line 117 and figure A2

Month/date was added when mentioning the DIN standards.

“ISO” was added when mentioning the 16610-49 standard.

On 3.3.1.1 “spherical-shaped” was corrected to “parabolical-shaped”

Round 2

Reviewer 2 Report

The revision made improved the paper. It is recommended for publication in its current version.